# Role of FGF-18 in Bone Regeneration

**DOI:** 10.3390/jfb14010036

**Published:** 2023-01-08

**Authors:** Kavipriya Murugaiyan, Sivashanmugam Amirthalingam, Nathaniel Suk-Yeon Hwang, Rangasamy Jayakumar

**Affiliations:** 1Polymeric Biomaterials Lab, School of Nanosciences and Molecular Medicine, Amrita Vishwa Vidyapeetham, Kochi 682041, India; 2Institute of Engineering Research, Seoul National University, Seoul 08826, Republic of Korea; 3School of Chemical and Biological Engineering, Institute of Chemical Processes, Seoul National University, Seoul 08826, Republic of Korea; 4BioMAX/N-Bio, Institute of BioEngineering, Seoul National University, Seoul 08826, Republic of Korea; 5Interdisciplinary Program in Bioengineering, Seoul National University, Seoul 08826, Republic of Korea

**Keywords:** fibroblasticgrowth factors (FGF), bone regeneration, osteogenesis, osseoconduction, osseoinduction, osseointegration, scaffolds

## Abstract

In tissue engineering, three key components are cells, biological/mechanical cues, and scaffolds. Biological cues are normally proteins such as growth factors and their derivatives, bioactive molecules, and the regulators of a gene. Numerous growth factors such as VEGF, FGF, and TGF-β are being studied and applied in different studies. The carriers used to release these growth factors also play an important role in their functioning. From the early part of the 1990s, more research has beenconductedon the role of fibroblast growth factors on the various physiological functions in our body. The fibroblast growth factor family contains 22 members. Fibroblast growth factors such as 2, 9, and 18 are mainly associated with the differentiation of osteoblasts and in bone regeneration. FGF-18 stimulates the PI3K/ERK pathway and smad1/5/8 pathway mediated via BMP-2 by blocking its antagonist, which is essential for bone formation. FGF-18 incorporated hydrogel and scaffolds had showed enhanced bone regeneration. This review highlights these functions and current trends using this growth factor and potential outcomes in the field of bone regeneration.

## 1. Introduction

Bone is a metabolically active organ comprising both organic and inorganic components with a plethora of functions to maintain homeostasis [1]. Bone loss happens in hereditary, deficiency, and pathological conditions as well as in trauma [2]; in these conditions, replacement certainly becomes vital for the survival of the patient. Autografts are regarded as the gold standard technique in the case of bone regeneration due to their aspects such as no risk of immunogenicity and their immediate availability from the donor, but the replacement of the donor site affects its morbidity, and restricted amounts of available graft make us reconsider the option. Allograft is an alternative option and is also greatly considered due to its ready availability and lack of requirement for a donor site. However, again it requires appropriate storage and sterilization techniques and carries a risk of immunogenicity [3]. Keeping in mind the complexity of the grafts and the structure, function, and composition of the bone itself, innovations are being conducted with new materials and methods to bio-mimic natural bone tissue.

Tissue engineering is an integrative field in which biological tissues are engineered accordingly to repair, replace, or regenerate lost tissues. This field uses a variety of materials ranging from natural to synthetic ones, and methods use various technologies to achieve its purpose [4]. The three important triads of tissue engineering are scaffolds, biological/mechanical cues, and cells [5]. Scaffolds are being developed to mimic the natural environment by binding certain bioactive molecules, while drugs are also being used to achieve this purpose [6]. Growth factors act either in an autocrine or in a paracrine manner and influence the cells around the scaffolds [7]. However, the problem arises when we expect regeneration for larger defects. Hence, synthetic bone grafts have been widely used in clinical scenarios due to their favorable properties to augment such large defects and stimulate bone regeneration [8,9]. Synthetic grafts can also be made osteoconductive/osteoinductive by the appropriate selection of graft materials [10]. The incorporation of fibroblastic growth factors (FGFs) has enhanced the performance of these synthetic grafts [11,12]. Fibroblastic growth factors play an important role in many physiological processes (i.e., inflammation, angiogenesis, etc.); one among these is skeletal development [2,8,13,14,15], where continuous bone remodeling happens in the presence of osteoblasts and osteoclasts ultimately necessitating the inevitable involvement of fibroblastic growth factors [16]. In the FGF family (Figure 1), FGF-2, -9, and -18 plays a significant role in bone regeneration [17]. Though in previous studies, more emphasis was laid on FGF-2, we here highlight the potential of FGF-18 when incorporated with hydrogel and scaffolds showing implicit bone regeneration.

## 2. Fibroblastic Growth Factors

The growth factors’ effects first came into light in 1939 after the mitogenic effect of the saline extract on chick was observed by Trowell and Willmer [18]. It was first purified and isolated later in the 1970s [19]. The FGFs are small, glycosylated proteins with a molecular weight of 17–34 kDa. Among mouse and humans, there are 22 FGF genes [20]. Fibroblastic growth factors (FGFs) are known for their potent mitogenic activity among a wide variety of cell types and developmental processes (Figure 2) [17,21].

FGFs of the paracrine family require cofactors such as heparin or heparan sulfate proteoglycans, whereas the FGF-endocrine family requires Klotho proteins as a co-receptor for their binding and initiating responses through FGF receptors (FGFR). Endocrine FGFs (FGF-15/19, FGF-21, and FGF-23) show negligible binding towards heparin and heparan sulfates, and they require α/β Klotho co-receptors to initiate their signaling activity. Heparin or heparan sulfate limits the diffusion of paracrine FGFs in the extracellular matrix (ECM), and due to the reduced affinity of endocrine FGFs, they freely diffuse through but require Klotho proteins for receptor binding [22]. Fibroblastic growth factors are divided into five paracrine sub-families, one intracrine sub-family, and one endocrine subfamily, based on their mechanism of action [17]. FGF-18 belongs to the FGF-8 subfamily. These growth factors perform their function by binding and activating FGF receptors (FGFR), i.e., FGFR-1, -2, -3, and -4, which come under the family of tyrosine kinase receptors. The FGF family, which is known for their activity in bone development are fibroblastic growth factors (FGF)-1, 2, 9, 10, and 18 [21]. Fibroblastic growth factors (FGFs) bind with fibroblastic growth factor receptors (FGFRs) and become activated by the phosphorylating tyrosine residues present in these receptors [23].

## 3. FGF-18 in Regeneration of Bone

FGF-18 was first reported in 1998 and was initially known for its activity in soft tissue proliferation, such as in the liver and intestine. The purified FGF-18 from mice contains 207 amino acids. FGF-18 is a glycoprotein with the first 26 amino acids forming a signal peptide sequence [24]. FGF-18 belongs to the paracrine FGF-8 subfamily. It exerts its action through FGFR-1, -2, and -3 (Figure 3) [25,26,27]. The gene which encodes for this protein is located in the fifth chromosome (5q35.1) [27,28]. The biological function of this growth factor is mainly through the proliferation of cells such as osteoblasts, chondrocytes, and osteoclasts [29].

Liu et al. demonstrated that FGF-18 acts as a ligand for FGFR-3 and mice lacking FGF-18 and showed delayed ossification and the reduction in osteogenic gene expression [14]. Ohbayashi et al. proved that FGF-18 is required for skeletal development by using an FGF-18^−/−^ deficient mouse model, which showed the delayed closure of sutures in calvarial bone [29]. Shimoka et al. also showed that FGF-18 may compensate for the role of FGF-2 on bone and cartilage since it is as equally potent as the latter [30]. FGF-18, along with other factors, regulates gene expression during skeletal development [31]. Recently, FGF-18 was immunostained in a developing fetal spine showing its importance in cellular proliferation and bone formation [32]. Recombinant FGF-18 improved the osseointegration of implants and prevented peri-implant fibrous response in FGFR-3^−/−^ mice [26]. Behr et al. showed that decreased FGF-18 affected the expression of RUNX2 and osteocalcin in FGF-18^+/−^ mice [33]. They also observed that the haploinsufficiency could not be compensated with other FGF ligands and bone morphogenic proteins (BMP)-2 in FGF-18^+/−^ mice. Wan et al. proved that the suppression of noggin-stimulated osteogenesis in vitro accelerated the same in vivo [34]. FGF-18-suppressed noggin is a BMP antagonist which positively augments the action of the bone morphogenic protein towards osteogenic differentiation [35]. BMP-2 is a potential osteoinductive growth factor [36], and FGF-18 further helps it in enhancing this property by inhibiting noggin [35].

In a study conducted by Hamidouche et al., the fact that FGF-18 promotes osteoblastic differentiation was experimentally confirmed by checking for osteoblastic markers, such as RUNX-2, ALP, and COL1A1 in C3H10T1/2, after treating them with the recombinant fibroblastic growth factor (rh FGF)-18. This was further corroborated by checking for alkaline phosphate (ALP) activity and in vitro matrix mineralization in the same cells. The authors displayed that FGF-18 enhanced osteogenic differentiation through the activation of the Extracellular signal-regulated kinase 1/2(ERK1/2) and Phosphatidylinositol 3-kinase (PI3K) pathway acting through FGFR-1 and 2 [25]. Fujioka-Kobayashi et al. studied the effect of different FGFs-8, 9, 17 and 18 on MC3T3-E1 cells, which are immature murine osteoblastic cells from mice, where FGF-18 alone with BMP-2 was sufficient to show significant osteogenic potential [37]. Behr et al. looked at the bone regeneration potential of FGFs-2, 9, and 18 in the cranial bone defects of adult mice. The results showed that FGF-18 was able to show higher healing potential compared to other growth factors (FGF-2 and 9). After the injury of the cranial defects, FGF-18^+/−^ mice showed no RUNX-2 expression, even though the weak expression was observed in fibroblastic growth factor (FGF)-9^+/−^ mice [38]. Jeon et al. showed the early osteogenic differentiation of FGF-18 on rat bone marrow-derived stem cells (BMSCs) by upregulating genes such as RUNX-2, Col1, and BMP-4 [39]. Nagayama et al. demonstrated that FGF-18 not only promotes osteogenic differentiation through BMP-2 but also supports the expression of FGFR-1, 2, and 3 in osteoblasts in the fetal coronal suture of mice [27]. A summary of the aforementioned pathways is given here (see Figure 4).

## 4. FGF-18 Incorporated Hydrogels, Scaffolds and Membranes in Bone Regeneration

FGF-18 has recently been incorporated into various scaffolds used for bone regeneration due to a plethora of studies substantiating its significance. FGF-18 incorporated hydrogels and scaffolds, more commonly used in bioceramics, in addition to it, therefore enhancing its potential and, in certain cases, controlling its release. FGF-18 was integrated into Chitin-Poly(lactic-*co*-glycolic acid) (PLGA) hydrogel along with calcium sulfate and checked for bone regeneration in vivo via cranial defect models in mice. This combination of FGF-18 showed good osteogenic effect compared to others that were used without growth factors. The bone volume was higher and showed a greater filling of the defect by the addition of FGF-18. It increased osteogenic differentiation through the increased expression of ALP. FGF-18 also increased the migration of HUVEC cells which is one of the requirements for effective bone regeneration [42]. In addition, FGF-18 was also checked for its synergistic/additive effect when used along with hydroxyapatite(HAP), bioglass(BG), and whitlockite(WH) nanoparticles incorporated into the Chitin-Polylactic-*co*-glycolic acid(PLGA) hydrogel(CG). Whitlockite nanoparticles incorporated in Chitin-Polylactic co-glycolic acid (PLGA) hydrogel(CGWH) showed a synergistic effect when combined with FGF-18 compared to Chitin-Polylactic co-glycolic acid(PLGA) hydrogel incorporated with Hydroxyapatite(CGHAP) and Bioglass(CGBG), which showed an additive effect in osteogenic gene expression and bone formation. Additionally, FGF-18 had an electrostatic interaction with the bio-ceramics, which led to the controlled release of FGF-18, thus further facilitating its use in bone regeneration [43]. The proposed mechanism of combining whitlockite nanoparticles and FGF-18 is shown in Figure 5. Fujioka-Kobayashi et al. studied the combination of FGF-18 with the bone morphogenic protein (BMP)-2 and found it was sufficient at showing significant bone regeneration potential in crosslinked acrylol cholestrol modified pullulan nanogels. These factors were combined with stimulated thick trabecular bone formation [37]. Charoenlarp et al. further proved the above findings by co-administering the two growth factors by developing a dried crosslinked nanogel (Nanoclik) and a dried porous crosslinked nanogel(NanoCliP). Both of them contained FGF-18 and BMP-2, and it was shown that Nanoclik showed better regeneration potential than NanoCliP [44]. Similarly, a hydrogel-based system containing mesoporous bioactive glass nanospheres loaded with FGF-18 was developed for inducing mesenchymal stem cells. Collagen gel incorporated with bioactive glass mesoporous nanocarriers loaded with the fibroblastic growth factor (FGF)-18 created a depot to support and signal mesenchymal stem cells (MSCs), showing increased osteogenesis within 1–2 weeks. It further upregulated bone morphogenic protein production, and the minerals from bioactive glass mimicked the natural bone tissue [45].

Core-shell nanofibrous scaffolds, with the capacity to deliver dual growth factors, were developed by Kang et al. [46], wherein mesoporous bioactive glass nanospheres(MBN) were used as a carrier to prolong the delivery of FGF-18 (for enhancing osteogenic differentiation) and FGF-2 (proliferation of cells and formation of blood vessels) was loaded freely in the electrospun mat for a rapid release profile. The nanofibrous scaffolds were made of poly(caprolactone) (outer part) and poly(ethylene oxide)(inner part), and the core contained the FGF-2 and FGF-18 loaded MBN. The regenerative potential of these electrospun nanofibrous scaffolds was observed in the rat calvarium defect model with evident bone formation [46]. A collagen membrane incorporated with fibroblastic growth factor (FGF)-18 showed effective osteogenic differentiation in MC3T3-E1 cell lines. This was due to the increased expression of RUNX-2 and Smad-5 and the downregulation of miR-133a and miR-135a. It provided the same osteoblastic activity compared to collagen membranes with a platelet-derived growth factor (PDGF) without cytotoxicity [47].

## 5. Future Perspectives and Conclusions

In this article, we have discussed generally fibroblastic growth factors and the role of FGF-18, specifically in bone regeneration and its significance. We also explained in detail the acceleration process of osteogenesis when FGF-18 was incorporated into the scaffolds. For clinical translation, FGF-18 faces problems similar to other growth factors, such as solubility, gradual diffusion, and paracrine action, and they also depend on the type and number of receptors, including intracellular events occurring after the binding of the growth factor and its ultimate ability to interact with the extra-cellular matrix (ECM). To circumvent these problems, new methods such as binding them covalently or non-covalently to the extracellular matrix (ECM) by modifying the growth factors sequence to include the ECM-binding domain and layer-by-layer assemblies of polyelectrolytes to retain the growth factors, in between them, the PEGylation of growth factors could improve the thermal stability and altering the protease-sensitive sites of these growth factors could be carried out. Another interesting fact is that biomaterials can be conjugated with heparan sulfate to mimic cell surface–ECM interactions with growth factors. Heparan sulfate is a linear polysaccharide that is expressed along with bone morphogenic proteins at the growth plate during skeletal development. The interaction of heparan sulfate and FGF-18 is studied [24] and their combined action on bone regeneration is one of the areas yet to be explored. We can also engineer exosomes for the delivery of FGF-18. Apart from this, we can also develop co-delivery or sequential delivery by designing carrier systems accordingly, opening the window for several possibilities to utilize FGF-18 in the area of bone regeneration. Sprifermin(recombinant FGF-18) underwent phase II clinical trials for osteoarthritis and showed an increase in cartilage thickness and provided symptomatic pain relief to the patient. Thus, we can also extend its application to bone tissue regeneration since we have promising results from the above-discussed studies [29,35,38]. With novel delivery systems, the release of the growth factor can be controlled in a precise manner and can be used in our favor. Therefore, more studies are encouraged to utilize FGF-18 for bone tissue engineering and thereby pushing it towards clinical translation.

## Figures and Tables

**Figure 1 jfb-14-00036-f001:**
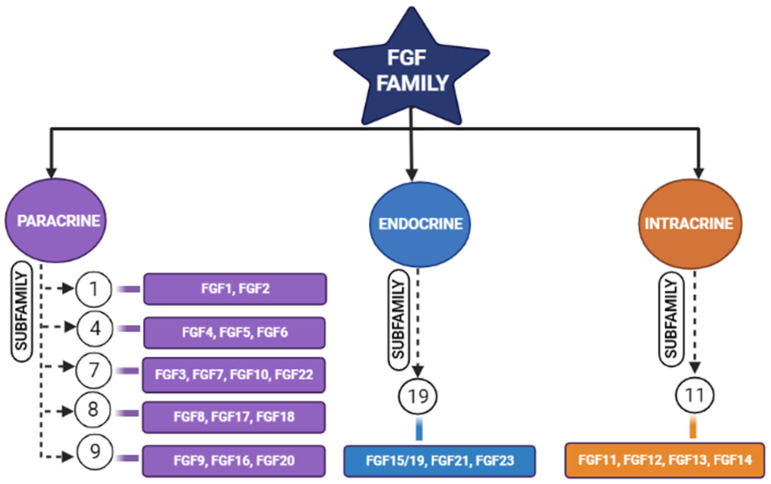
Schematic diagram representing different members of the fibroblastic growth factors (FGF) family and their subfamilies [17].

**Figure 2 jfb-14-00036-f002:**
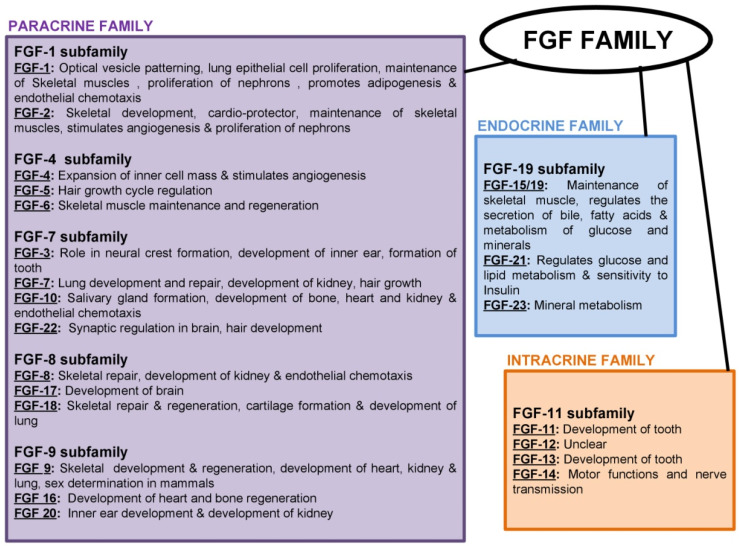
Schematic diagram showing the functions of FGF family [17,21].

**Figure 3 jfb-14-00036-f003:**
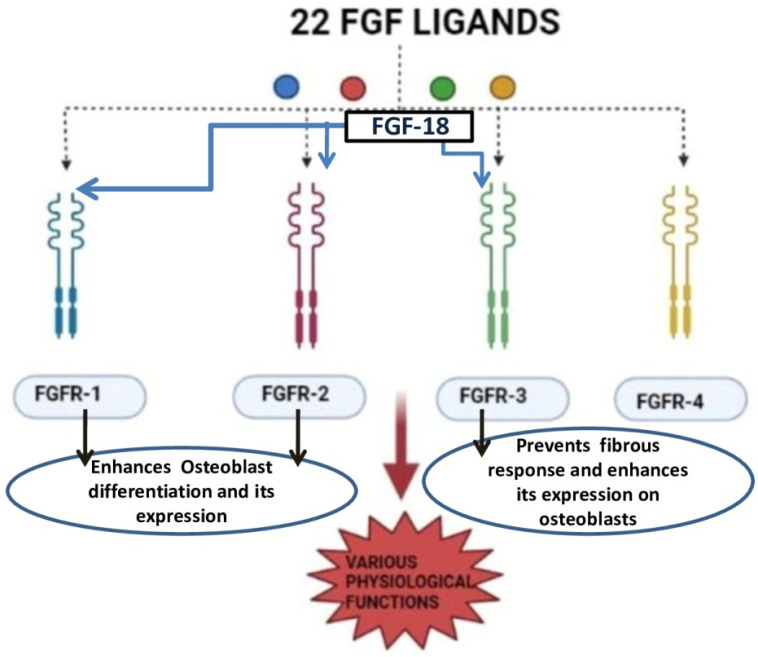
Diagram represents different types of receptors of the FGF family and FGF-18 action on selective receptors [25,26,27].

**Figure 4 jfb-14-00036-f004:**
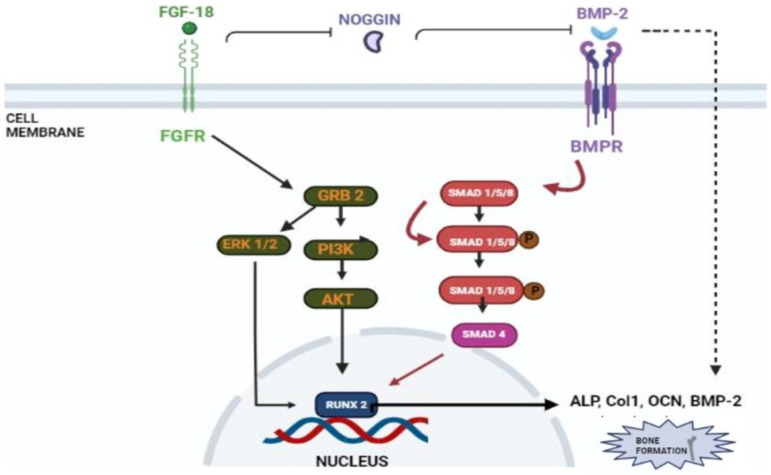
Schematic representation of FGF-18’s role in osteogenic differentiation. FGF-18 inhibits noggin which is a BMP-2 antagonist [33]. BMP activates the smad 1/5/8 pathway which activates early osteoblastic markers such as ALP, Col1, OCN and RUNX-2 [40,41]. FGF-18 themselves activate PI3K-ERK pathway which also activates the RUNX-2 gene [39] which is an important marker for osteogenic differentiation.

**Figure 5 jfb-14-00036-f005:**
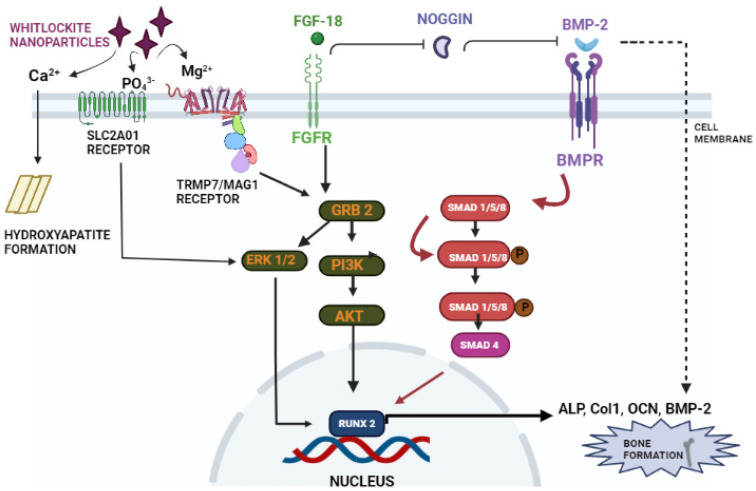
Schematic diagram representing the proposed mechanism of combining whitlockite nanoparticles and FGF-18 for bone regeneration.

## Data Availability

Not applicable.

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
