# Peer review of "Role of FGF-18 in Bone Regeneration"

_jfb, 2023, doi:10.3390/jfb14010036_

Round 1

Reviewer 1 Report

Thanks to the editor.

The paper is well organized and the topic is quite interesting.

I suggest to check spacing and or spelling mistakes.

Author Response

Answer: Thank you for your valuable comment. We have corrected the spacing & spelling mistakes in the revised manuscript as per your suggestion. The revised part is highlighted in yellow colour.

Reviewer 2 Report

Dear Authors

The editorial part of the work deviates from the accepted format. Both the text and drawings require a thorough reworking and adaptation to the applicable pattern.

The work is very interesting, however, the authors' own research presented in a descriptive form supported by the literature on the subject does not give a full picture of the novelties introduced by the authors and original innovative solutions. I suggest you give more prominence to your own achievements, and clearly define why your research is so unique that it is worth presenting it in such a reputable journal. The reader should clearly deduce which part of the work concerns the review of literature and which part is the authorial, innovative achievement of the authors. I suggest that some of the authors' own research along with the presentation of specific research results should be presented as a separate point in which you will not refer to the research of other authors (literature review), but will unambiguously present your own research results on the basis of relevant documentation.

The presented research (including a very thorough and substantive literature review) is thematically coherent and is confirmed by modern research trends of world science, but the work submitted for review lacks a clear presentation, which is a unique and documented achievement of the authors. What is the innovation and "superiority" of your research over such widely published works of other authors from other research centers (including those cited by you).

Author Response

Thank you for your valuable comment. The aim of this review article is to highlight only the role FGF-18 in bone regeneration. We have quoted all the published works which are related to FGF-18 in order to substantiate the potential of the growth factor which is often been overlooked. FGF-18 is the only growth factor based drug to enter into Clinical trials for osteoarthritis. We have included the same is in the revised manuscript (Ref: The effects of sprifermin on symptoms and structure in a subgroup at risk of progression in the FORWARD knee osteoarthritis trial by Guering et al). There are a limiting number of studies implying its role in bone regeneration even though its efficacy has been superior than the others shown in previous studies.

The revised part is highlighted in yellow color.

Reviewer 3 Report

Author clarified the role of FGF-18 in bone regeneration alternative to bone grafting such as allograft and autograft. Authors indicated that incorporation of FGF enhanced the properties of the synthetic grafts. However, some information should be indicated.

Major comments

1.     Authors presented that FGF-2, 9, and 18 play a crucial role in bone regeneration whether they belong to different subfamily. In addition, author indicated that study of FGF-2 on bone regeneration is more emphasis. However, authors did not explain well why FGF-18 should be focused on FGF-18 rather that FGF-2 and FGF-9.

2.     Authors focused on FGF-18 in bone regeneration. However, the summarized table of the function of 22 FGF should be added.

3.     Authors presented the incorporation between FGF-18 and hydrogels and scaffold. To emphasize an idea of FGF-18 plays an important role in bone regeneration, other types of FGF should indicated.

Minor comments

1.     Abbreviations should be used after the full name and be used through the article.

2.     Space and full stop must be checked through the manuscript following the format of JFB.

3.     The typing, such as et al., or et al should follow the format of JFB.

4.     The capital letter should be checked through the text such as Noggin or noggin, Mesenchymal or mesenchymal.

5.     Some references are appropriate but should be updated.

6.     Line 138, typing must be checked.

Author Response

Comments and Suggestions for Authors. Author clarified the role of FGF-18 in bone regeneration alternative to bone grafting such as allograft and autograft. Authors indicated that incorporation of FGF enhanced the properties of the synthetic grafts. However, some information should be indicated.

Major comments

  1. Authors presented that FGF-2, 9, and 18 play a crucial role in bone regeneration whether they belong to different subfamily. In addition, author indicated that study of FGF-2 on bone regeneration is more emphasis. However, authors did not explain well why FGF-18 should be focused on FGF-18 rather that FGF-2 and FGF-9.

Answer: The third heading Fibroblastic growth factor-18 in regeneration of bone explains why FGF-18 should be given more attention compared to FGF-2 and FGF-9. We have included the relevant following references in the revised manuscript.

1) Regulation of Osteoblast, Chondrocyte, and Osteoclast Functions by Fibroblast Growth Factor (FGF)-18 in Comparison with FGF-2 and FGF-10 by Shimoka et al [29].

2) Different endogenous threshold levels of Fibroblast Growth Factor-ligands determine the healing potential of frontal and parietal bones by Behr et al [37].

  1. Authors focused on FGF-18 in bone regeneration. However, the summarized table of the function of 22 FGF should be added.

Answer: We have included Diagram 2, which now contains all the Functions of the FGF family.

  1. Authors presented the incorporation between FGF-18 and hydrogels and scaffold. To emphasize an idea of FGF-18 plays an important role in bone regeneration, other types of FGF should indicated.

Answer: In our future review, we will include the hydrogels or scaffolds developed using all Fibroblastic growth factors.

Minor comments

  1. Abbreviations should be used after the full name and be used through the article.
  2. Space and full stop must be checked through the manuscript following the format of JFB.
  3. The typing, such as et al., or et al should follow the format of JFB.
  4. The capital letter should be checked through the text such as Noggin or noggin, Mesenchymal or mesenchymal.
  5. Some references are appropriate but should be updated.
  6. Line 138, typing must be checked.

Answer:The above mentioned minor comments have been corrected in the revised manuscript.

The revised parts are highlighted in yellow color.

Reviewer 4 Report

The review is devoted to the role of FGF-18, one of the members of a large family of FGF, in bone tissue regeneration. The review is well written, illustrated and provides insight into the role of FGF-18 in bone formation. The paper provides very interesting data but it still needs a several revision. The authors do not mention that FGF-18, like all members of the FGF family, belong to heparin-binding factors. FGFs can bind to FGFRs with the help of heparan sulfate, a co-factor, and thereby induce their biological effects through activation of four major signaling pathways. FGFs bind to FGFRs through a heparan sulphate glycosaminoglycan binding site, limiting their diffusion through the ECM. All effects of FGF-18 a close resemblance to those of FGF-2. Activation of the MAPK pathway, in response to FGF-18 signaling, is key in determining the activity of RUNX-2, a master transcription factor of bone formation. To obtain optimal results in vivo, it is important to improve the half-life of FGF-18 and its biological stability. I hope these comments will be helpful.

Author Response

Thank you for your valuable comments. We have also added them under the second heading Fibroblastic growth factors.

The revised part is highlighted in yellow color.

Reviewer 5 Report

The authors have written a manuscript about the role and pathway of FGFs, and the possible application of them. The actual FGF-18 description is actually too short, and the general FGF family description is too long. Thus, the specifity of the title is not justified in the text.

Author Response

Thank you for your valuable comment. We have included the description of FGF-18 as per your suggestion with available literature.

The revised part is highlighted in yellow color.

Round 2

Reviewer 3 Report

Authors should check spacing throughout the manuscript. There are still mistakes. Consistent formatting should be used throughout.

Reviewer 5 Report

The manuscript has been updated with relevant information.